# Puerarin Induces Molecular Details of Ferroptosis-Associated Anti-Inflammatory on RAW264.7 Macrophages

**DOI:** 10.3390/metabo12070653

**Published:** 2022-07-15

**Authors:** Jinzi Zeng, Ning Zhao, Jiajia Yang, Weiyang Kuang, Xuewei Xia, Xiaodan Chen, Zhiyuan Liu, Riming Huang

**Affiliations:** 1Guangdong Provincial Key Laboratory of Food Quality and Safety, College of Food Science, South China Agricultural University, Guangzhou 510642, China; zengjinzi1999@stu.scau.edu.cn (J.Z.); yangjiajia@stu.scau.edu.cn (J.Y.); 20212145014@stu.scau.edu.cn (W.K.); xiaxuewei@stu.scau.edu.cn (X.X.); 18125907913@163.com (X.C.); aiden@stu.scau.edu.cn (Z.L.); 2Graduate School, Guangzhou University of Chinese Medicine, Guangzhou 510006, China; 20211111548@stu.gzucm.edu.cn

**Keywords:** puerarin, anti-inflammatory, ferroptosis, network pharmacology, metabolomics, molecular mechanism

## Abstract

Puerarin is a natural flavonoid with significant anti-inflammatory effects. Recent studies have suggested that ferroptosis may involve puerarin countering inflammation. However, the mechanism of ferroptosis mediated by the anti-inflammatory process of puerarin has not been widely explored. Herein, puerarin at a concentration of 40 μM showed an anti-inflammatory effect on lipopolysaccharide (LPS)-induced macrophages RAW264.7. The analysis of network pharmacology indicated that 51 common targets were enriched in 136 pathways, and most of the pathways were associated with ferroptosis. Subsequently, the analysis of metabolomics obtained 61 differential metabolites that were enriched in 30 metabolic pathways. Furthermore, integrated network pharmacology and metabolomics revealed that puerarin exerted an excellent effect on anti-inflammatory in RAW264.7 via regulating ferroptosis-related arachidonic acid metabolism, tryptophan metabolism, and glutathione metabolism pathways, and metabolites such as 20-hydroxyeicosatetraenoic acid (20-HETE), serotonin, kynurenine, oxidized glutathione (GSSG), gamma-glutamylcysteine and cysteinylglycine were involved. In addition, the possible active binding sites of the potential targeted proteins such as acyl-CoA synthetase long-chain family member 4 (ACSL4), prostaglandin-endoperoxide synthase 2 (PTGS2), arachidonate 15-lipoxygenase (ALOX15) and glutathione peroxidase 4 (GPX4) with puerarin were further revealed by molecular docking. Thus, we suggested that ferroptosis mediated the anti-inflammatory effects of puerarin in macrophages RAW264.7 induced by LPS.

## 1. Introduction

Inflammation is a kind of immune defense response to harmful stimulation, which plays an important role in self-renewal [1]. However, uncontrolled inflammation is implicated in multiple diseases such as cancer, neurodegenerative diseases, cardiovascular diseases, and diabetes [2]. Therefore, researchers have always devoted themselves to understanding the mechanism of inflammation and finding ways to eliminate excessive inflammatory responses. In recent years, studies have demonstrated that non-apoptotic death had immunogenicity. Ferroptosis, as a new form of non-apoptotic cell death with immunogenicity, strongly aroused researchers’ interest [3]. It is characterized by iron overloading, the expression of the cystine/glutamate antiporter, reactive oxygen species (ROS) accumulation, and lipid peroxidation [4]. Interestingly, some researchers found a potential association between ferroptosis and inflammation. According to Wang et al. [4], when ferroptosis occurs in cells, inflammation-linked, damage-related molecules will be released and then recognized by immune cells to trigger the inflammatory response. The accumulation of ROS is a major cause of ferroptosis, which also can be the inducement of the inflammatory cytokine outbreak [5,6]. Studies showed that the large production of ROS and oxidative stress could activate the NF-κB signaling pathway, which could exacerbate the inflammatory response [7]. Meanwhile, the increased levels of oxygenates such as lipoxygenases (LOXs) and PTGS2 during ferroptosis result in a large number of lipid peroxidation products, which modulate the immune response [8]. In addition, GPX4 is a key inhibitor of ferroptosis, which is also an essential participant in inflammation suppression. It can inhibit inflammation by reducing hydroperoxyl groups of complex lipids and silencing lipoxygenases [9]. Thus, ferroptosis is thought to be associated with inflammation, but the detailed connection between them is still unclear.

*Pueraria lobata* is not only a traditional Chinese herbal medicine used for thousands of years but is also used as a dietary supplement in southern China. Puerarin, as a natural flavonoid compound isolated from the root of *Pueraria lobata,* has been shown to have many bioactive functions (anti-inflammatory, antioxidant and antiviral [10]). Especially, previous studies showed that puerarin has a significant effect on the treatment of inflammatory diseases, such as cancer [11], diabetes [12], Alzheimer’s disease [13] and cardiovascular disease [14], which evoked the interest of researchers. At present, there are many studies on the anti-inflammatory activity of puerarin. The anti-inflammatory effects of puerarin mainly focus on the following classical ways of affecting the immune cells (by inhibiting the differentiation and increment of T lymphocytes and improving macro-phage phagocytosis and so on), inflammatory factors (by inhibiting proinflammatory factors and promoting anti-inflammatory factors and so on) and signal pathways (by inhibiting the activation of NF-κB signaling pathway and MAPK signaling pathway and so on) [15]. Interestingly, some recent research showed that puerarin-alleviated inflammatory diseases are often accompanied by the inhibition of ferroptosis. Xu et al. [16] found that puerarin could inhibit ferroptosis and inflammation by alleviating lipid peroxidation (GSH, MDA and ROS), downregulating the levels of pro-inflammatory factors (IL-6, TNF-α and IL-1β) and regulating the levels of ferroptosis related proteins (SLC7A11, GPX4, FTH1 and NOX1). What is more, animal studies have found that puerarin could protect against heart failure induced by pressure overload via alleviating ferroptosis and Nox4 signaling might be involved in this process [17]. Zhou et al. [18] also found that AMPK-mediated ferroptosis signaling was involved in the course of puerarin countering sepsis-induced myocardial injury, and the AMPK signaling pathway was supposed to be highly related to inflammation [19]. In addition, as a natural compound with antioxidant activity, puerarin can improve lipid metabolism disorder by inhibiting oxidative stress [20], which also shows the potential to resist inflammation and ferroptosis. In general, current studies hinted that the anti-ferroptosis effect might involve the anti-inflammatory process of puerarin. Although some reports support this hypothesis, it is still not sufficiently demonstrated that ferroptosis mediated the anti-inflammatory effect of puerarin, and their detailed mechanisms still need to be further explored.

In this study, the RAW264.7 was induced by LPS and the anti-inflammatory effect of puerarin was evaluated by the release of NO, IL-6 and TNF-α. Furthermore, network pharmacology and metabolomics analysis were used to clarify the possible anti-inflammatory mechanism related to the ferroptosis of puerarin. Finally, the key proteins associated with ferroptosis were bound to puerarin by molecular docking to verify that ferroptosis mediated the anti-inflammatory effect of puerarin.

## 2. Results

### 2.1. The Anti-Inflammatory Effect of Puerarin on RAW264.7

The cell viability of RAW264.7 cells is determined by the cell-counting kit-8 (CCK-8) assay, non-treated as the control group. As shown in Figure 1A, the cell viability of RAW264.7 cells was not significantly changed by incubating with puerarin concentrations of 10, 20, 40 and 80 μM for 24 h, which also indicated that puerarin was not cytotoxic to RAW264.7 cells within this concentration range. In addition, the stimulation of RAW 264.7 with LPS (2.5 μg/mL) for 24 h resulted in a significant increase in NO, IL-6 and TNF-α compared with the control group. Although pretreatment with puerarin (at the concentrations range of 10–80 μM) did not reduce the production of NO back to normal as the control group, puerarin at 40 μM reduced the release of NO to some extent (Figure 1B). Meanwhile, regarding IL-6 and TNF-α as pro-inflammatory cytokines, their production was significantly decreased by pretreatment with puerarin (40 μM) (Figure 1C,D). All the results show that puerarin could relieve inflammation at a concentration of 40 μM. Hence, the following cell experiments were carried out under this condition.

### 2.2. Result of Network Pharmacology

According to Figure 2A, 108 puerarin-related targets and 2800 inflammation-related targets were obtained from the databases, and 78 common targets were identified by the Venn diagram. After screening by the STRING database, 51 of 78 targets were launched into protein–protein interaction (PPI) networks (Appendix A). According to the principle that the degree value is greater than the median (degree = 6), 18 targets are considered to play an important role in the anti-inflammatory process of puerarin (Table 1). Moreover, we found that some targets (such as *Ahr*, *Ptgs2* and *Alox15*) highly correlated with ferroptosis [21,22,23]. The outcomes of the Kyoto Encyclopedia of Genes and Genomes (KEGG) analysis (Figure 2B, Appendix A) suggested that the top 20 pathways could be roughly divided into inflammation-related, cancer-related and virus-related. Among the majority of pathways, they are highly associated with inflammation, especially the TNF signaling pathway [24] and the IL-17 signaling pathway [25]. In addition, ferroptosis was thought to be related to the anti-inflammatory function of puerarin owing to these pathways, which included pathways in cancer [26], lipid and atherosclerosis [27], apoptosis [28], colorectal cancer [29], tuberculosis [30], and prostate cancer [31]. Based on the result of network pharmacology, we speculated that ferroptosis is involved in the anti-inflammatory process of puerarin. However, this speculation still needs to be further proved.

### 2.3. Result of Metabolites

The results of the multivariate data analysis demonstrated that the model of inflammation was successfully established and was good for prediction (Appendix A). According to VIP > 1 and *p* < 0.1, 61 differential metabolites were screened between the LPS group and the puerarin group, of which 6 were up-regulation and 55 were down-regulation (Figure 3A,B). Subsequently, these 61 differential metabolites were found to become enriched in 30 KEGG pathways (Figure 3C, Appendix A), and these pathways were mainly concentrated in amino acid metabolism, carbohydrate and energy metabolism, and lipid metabolism (Figure 3D). Amino acid metabolism was proved to be highly associated with inflammation and ferroptosis due to its functions and crucial role in signaling pathways, such as tryptophan in inflammation and cystine in ferroptosis [32]. Additionally, energy stress can mediate the occurrence of ferroptosis and inflammation because these processes are accompanied by the massive consumption of energy [33,34]. Lipid peroxidation can result from lipid metabolism disorders, which is also the inducement of ferroptosis and inflammation [35]. Based on the result, we speculated that puerarin had an anti-inflammatory effect by mediating amino acid metabolism, carbohydrate and energy metabolism, and lipid metabolism.

### 2.4. Combined Analysis of Metabolomics and Network Pharmacology

To explore the crucial metabolic pathways, 136 KEGG pathways enriched by network pharmacology and 30 KEGG pathways enriched by metabolomics were intersected. Finally, we obtained 3 pathways: arginine and proline metabolism, arginine and proline metabolism tryptophan metabolism, and arachidonic acid metabolism (Figure 4A). Among these 3 pathways, arginine and proline metabolism was thought to be highly connected to inflammation, causing the nitric oxide synthases and the arginase enzymes in the immune response [36]. Then, we learned that the tryptophan metabolism and arachidonic acid metabolism are closely related to inflammation and ferroptosis [37,38,39]. In addition, the glutathione metabolism pathway, as a key pathway related to ferroptosis, is also enriched in metabolomics. Whereupon, we next paid close attention to arachidonic acid metabolism, tryptophan metabolism and glutathione metabolism, which are involved in inflammation and ferroptosis. According to Figure 4B, 6 targets obtained from the network pharmacology analysis and 10 reversed metabolites gathered by metabolites analysis in these 3 pathways were represented. Significantly, PTGS2 and ALOX15 are two important enzymes in the arachidonic acid metabolic pathway, which are closely associated with lipid peroxidation, immune enhancement and ferroptosis [40]. Meanwhile, the expression of 20-HETE, kynurenine and serotonin decreased, while the expression of oxidized glutathione (GSSG), gamma-glutamylcysteine and cysteinylglycine increased, which showed that the occurrence and development of inflammation and ferroptosis were modulated [38,40]. Therefore, we concluded that puerarin played an anti-inflammatory role through tryptophan metabolism, arachidonic acid metabolism and glutathione metabolism, and ferroptosis was extremely likely involved.

### 2.5. Molecular Docking

Molecular docking was used to verify whether puerarin can directly participate in the regulation of ACSL4, PTGS2, ALOX15 and GPX4 or not. Among them, ACSL4, PTGS2 and GPX4 are the biomarkers of ferroptosis [41] and ALOX15 is related to the occurrence of ferroptosis [42]. As shown in Table 2 and Figure 5, we found that puerarin could bind to these 4 protein receptors mainly by forming a strong hydrogen-bonding interaction. In addition, the Pi–Pi interaction and Pi–cation interaction were also present in puerarin with TYR466 containing benzene and ARG570 bearing cation in the binding mode of puerarin and ACSL4, respectively. TYR355 in PTGS2 could form Pi–Pi interaction with puerarin as well. According to the prediction of prime MM/GBSA, the binding free energies of puerarin to ACSL4, PTGS2, ALOX15 and GPX4 are −55.41, −24.62, −27.73 and −17.37 kcal/mol (Figure 5), separately. Thus, we infer that puerarin acts as an anti-inflammatory by acting directly on the above proteins.

## 3. Discussion

As we all know, inflammation is often accompanied by the production of small molecular inflammatory mediators and the expression of inflammation-related enzymes at the molecular level. As the cell wall components of Gram-negative bacteria, LPS can be the pathogen-associated molecule recognized by immune cells to stimulate innate immunity, which has also been widely used for modeling inflammation [43]. When LPS bound to TLRs receptors on the surface of macrophages RAW264.7, many inflammatory mediators were released soon afterward [18]. According to the above results, the LPS-induced inflammatory response could be reversed by puerarin in RAW264.7 because of the down-regulation of NO, IL-6 and TNF-α. The studies of Xu et al. [16] and Zhou et al. [18] also showed the same effect.

Additionally, the intracellular metabolic disorder is also a sign of inflammation, which included lipid peroxidation; protein degradation; and the metabolic disorder of amino acid, carbohydrate and energy metabolism abnormality [44]. Meanwhile, many metabolics involved in iron, lipids and amino acids were thought to participate in the regulation of ferroptosis via regulating iron accumulation or lipid peroxidation directly or indirectly [8]. The combined analysis of metabolomics and network pharmacology showed that puerarin might play an anti-inflammatory role through the pathways of arachidonic acid metabolism, tryptophan metabolism and glutathione metabolism. Interestingly, these pathways are considered to be both involved in inflammation and ferroptosis. Therefore, we put forward that one of the anti-inflammatory pathways of puerarin might mediate ferroptosis by regulating arachidonic acid metabolism, tryptophan metabolism and glutathione metabolism. The probable mechanism of puerarin countering inflammation is shown in Figure 6.

Arachidonic acid usually exists in the cell membrane in the form of phospholipid. When the cell is under stress, it is released from the phospholipid in the form of free arachidonic acid [45]. ACSL4 is an important enzyme in eicosanoid biosynthesis, which can catalyze the addition of CoA to the long-chain polyunsaturated bonds of arachidonic acid involved in the further inflammatory process [8]. In addition, ACSL4 is also a biomarker of ferroptosis because it contributes to cell ferroptosis by promoting lipid peroxidation [46]. PTGS2 and ALOX15 are enzymes mainly involved in arachidonic acid metabolism, in which ALOX15 initiates ferroptosis by oxidizing PUFA-PE and PTGS2 is markedly up-regulated during ferroptosis [45]. Our investigation showed that puerarin could bind to ACSL4, PTGS2 and ALOX15 with low free binding energy. Whereupon, we suggested that puerarin reduced lipid peroxidation and ferroptosis to ease inflammation by suppressing the activation of the above protein in arachidonic acid metabolism. At the same time, some studies demonstrated that puerarin could inhibit the expression of ACSL4 and PTGS2 and then reduce lipid peroxidation and regulate lipid metabolism disorder [16,18,47]. Moreover, ALOX15 was found to be downregulated after treatment with flavonoids such as puerarin [48]. More importantly, the products of arachidonic acid metabolism are inflammatory mediators such as prostaglandins, leukotrienes, epoxyeicosatrienoic acids and hydroxyeicosatetraenoic acids [45]. They will break the balance of lipid metabolism in cells, leading to lipid peroxidation, and then produce a large number of ROS to further exacerbate the inflammatory response [49]. 20-HETE, as a lipid peroxidation metabolite, is believed to promote the production of ROS and protein peroxidation [50]. The reduction of 20-HETE compared with the LPS group in our study reflected inflammation remission by puerarin. Thus, we suggested that puerarin be used to counter inflammation via anti-ferroptosis by regulating arachidonic acid metabolism.

Tryptophan metabolism is highly correlated with the function of macrophages, and its increased decomposition has been observed in inflammation [51]. However, the depletion of tryptophan and the production of metabolites are complex. Tryptophan is an essential amino acid and a precursor of serotonin, kynurenine and indole [52]. Serotonin is widely believed to be a neuromodulator, and it is also a mediator of inflammation. It can promote the release of cytokines by activating the NF-κB signaling pathway [52,53]. The downregulation of serotonin in metabolomics analysis suggested that puerarin prevented tryptophan from being metabolized into serotonin to reduce inflammation. Furthermore, the tryptophan-kynurenine pathway plays an important role in inflammation and ferroptosis [39,40]. When cells are in a state of inflammation, some cytokines or the consumption of tryptophan can activate IDO1, then tryptophan is metabolized to kynurenine [38,54]. Kynurenine can induce the activation of AHR, the expression of CYP1A1, and the generation of ROS and (eventually) cell ferroptosis [55]. Compared with the LPS group, the expression of kynurenine was down-regulated after pretreatment with puerarin. In addition, flavonoids similar to puerarin, myricetin and flavonoids isolated from *Sophora flavescens* could suppress the expression of IDO1. [56,57]. So, we believed that the tryptophan-kynurenine metabolic pathway was delayed when puerarin entered the cell, which demonstrated that ferroptosis and inflammation were under control. Thus, by regulating ferroptosis-related tryptophan metabolism, puerarin could adjust the status of inflammation.

As for glutathione metabolism, we know the importance of maintaining the balance of ROS and intracellular redox homeostasis in the whole inflammatory response through the previous introduction. Glutathione is the most important antioxidant in the human body, which can eliminate ROS and protect cells from oxidative stress [58]. Oxidized glutathione GSSG is the oxidized form of glutathione, which also can be used by the body and converted into reduced glutathione in the presence of NADPH [59]. In addition, cysteinylglycine and gamma-glutamylcysteine are important metabolites in the glutathione metabolic pathway because gamma-glutamylcysteine is the immediate precursor of GSH with ROS detoxification ability, and gamma-glutamylcysteine is composed by cysteinylglycine and glutamate [60]. In our study, it was shown that pretreatment with puerarin increased the levels of GSSG, cysteinylglycine and gamma-glutamylcysteine in LPS-induced RAW264.7. Moreover, puerarin was found to up-regulate the expression of GSH [16,18]. Hence, we supposed that puerarin could promote the biosynthesis of GSH to inhibit ferroptosis, then relieve inflammation. As an antioxidant defense enzyme in glutathione metabolism, GPX4 can regulate eicosanoid biosynthesis and ferroptosis, and its upregulation can efficiently suppress inflammation and ferroptosis [16,61,62]. Many studies have shown that the level of GPX4 increases following the administration of puerarin. Moreover, Huang et al. [63] reported that compound 1d4 can be an allosteric activator to activate GPX4 for connecting with the key allosteric site (D21 and D23) and key catalytic residues (C46, Q81, K90, W136 and so on). Meanwhile, the result of molecular docking predicted that puerarin may bind to GPX4 with residues ASP21, ASP23, LYS90 and ASP101. For that, we inferred puerarin may be the potential allosteric activator of GPX4. In short, we believed that puerarin activated the glutathione metabolism to block ferroptosis and further affected inflammation.

In fact, in addition to puerarin, a similar effect also occurred in many other flavonoid compounds, such as quercetin [64], baicalin [65,66] and carthamin yellow [67]. All of them possess the abilities of anti-inflammation and anti-ferroptosis. Wang et al. [64] found that quercetin ameliorates acute kidney injury through the inhibition of ferroptosis and following inflammation. Additionally, the expression of GSH, GPX4 and SC7A11 in the glutathione metabolism pathway increased after treatment with quercetin. In addition, protein ACSL4 in the arachidonic acid metabolism pathway was down-regulation by baicalin to inhibit ferroptosis and then prevent myocardial ischemia/reperfusion (I/R) injury [66]. Therefore, we think it is credible that puerarin inhibits inflammation by regulating ferroptosis.

## 4. Materials and Methods

### 4.1. Materials

Puerarin was purchased from Solarbio Biotechnology Co., Ltd. (Beijing, China). Macrophage RAW 264.7 cells were obtained from Jinan University (Guangzhou, China). Lipopolysaccharide (LPS) was purchased from Shanghai Yuanye Biotechnology Co., Ltd. (Shanghai, China). Dulbecco modified eagle medium (DMEM), phosphate buffer solution (PBS), penicillin-streptomycin and fetal bovine serum (FBS) were supplied by Gibco Life Technologies (Grand Island, NY, USA). The nitric oxide (NO) kit and cell counting kit-8 (CCK-8 kit) were purchased from Beyotime Biotechnology Co., Ltd. (Shanghai, China) and Data Invention Biotechnology Co., Ltd. (Guangzhou, China), respectively. Mouse IL-6 and TNF-α ELISA kits were purchased from CUSABIO Biotechnology Co., Ltd. (Wuhan, China). In this study, all other reagents were analytical grade.

### 4.2. Cell Culture

RAW264.7 cells were cultured at 37 °C in a humidified atmosphere containing 5% CO_2_. DMEM supplemented with 10% (*v*/*v*) FBS and 1% (*v*/*v*) penicillin-streptomycin was used as cell medium. Additionally, RAW264.7 was cultivated in sterile tissue culture flasks and cultured for further experimentation.

### 4.3. Detection of Cell Viability

Cell viability was measured using the cell counting kit-8 (CCK-8) assay. RAW264.7 cells were seeded into 96-well (100 μL/well) plates at a density of 1 × 10^4^ cells per well. When the cells were in the logarithmic phase, they were exposed to different concentrations of puerarin (10, 20, 40 and 80 μM) for 24 h. The non-treated RAW 264.7 cells were used as the control. After incubating for 24 h, 10 μL of CCK-8 solution was added to each well and incubated for 2 h. The absorbance of each well was measured by a microplate reader at 450 nm. For cell viability assay, three independent experiments were carried out and four replicates per experimented on.

### 4.4. Detection of NO and Cytokine Production

The levels of NO, IL-6 and TNF-α in the supernatants of the RAW264.7 cells from each group were determined using commercial kits following the manufacturer’s instructions. Briefly, RAW264.7 cells were seeded in 12-well (2 mL/well) plates at 2 × 10^5^ cells/well and cultured for 24 h. Subsequently, RAW264.7 cells were incubated with puerarin at different concentrations (0, 10, 20 and 40 μM) and then treated with LPS (2.5 μg/mL). Puerarin was added 30 min prior to LPS, and non-treated and LPS only-treated RAW 264.7 cells were used as the control group and LPS group, respectively. For another 24 h, cell supernatant was collected for the detection of NO, IL-6 and TNF-α. All experiments were conducted three times, with three biological replicates per concentration.

### 4.5. Network Pharmacology Analysis

The integrative pharmacology-based research platform of traditional Chinese medicine (TCMIP) [68], traditional Chinese medicine systems pharmacology (TCMSP) [69], the Swiss target prediction [70] and the TargetNet database [71] were used for screening puerarin-related targets. The pharmacogenetics and pharmacogenomics knowledge base (PharmGkb) [72], GeneCards [73], Online Mendelian Inheritance in Man (OMIM) [74], comparative toxicogenomics database (CTD) [75] and therapeutic target database (TTD) [76] were used for screening inflammation-related targets with keywords “inflammation”. Venn diagram was created by ggVenn (R package) to show the common targets. The protein–protein interaction network (PPI) model with species set to “Mus musculus [10090]”, the minimum required interaction score set to “highest confidence (0.900)” and the “hide disconnected nodes in the network” were constructed by STRING11.5 [77] and Cytoscape3.9.0 software. Finally, we used the Database for Annotation, Visualization, and Integrated Discovery (DAVID 6.8) [78] to analyze the Kyoto Encyclopedia of Genes and Genomes (KEGG) pathway. The above databases were accessed on 6 January 2022.

### 4.6. Metabolites Analysis

The cell samples (three biological replicates in each group) for metabolite analysis, including the control, LPS-, and puerarin (40 μM)-treated groups of the macrophage RAW264.7 cells, were carried out with a standard metabolic operating procedure [79]. Then, the Vanquish UHPLC System (Thermo Fisher Scientific, Waltham, MA, USA) and Orbitrap Exploris 120 (Thermo Fisher Scientific, Waltham, MA, USA) with ESI ion source were used to acquire LC-MS raw data. The samples were performed chromatographic separation on an ACQUITY UPLC ^®^ HSS T3 (150 × 2.1 mm, 1.8 μm) (Waters, Milford, MA, USA). The mobile phase system and gradient elution refer to Zelena [80] and Want [81]. Subsequently, principal component analysis (PCA), partial least-square discriminant analysis (PLS-DA), and orthogonal partial least-square discriminant analysis (OPLS-DA) were applied to data analysis. Finally, *p* < 0.1 and VIP > 1 were considered to be differently accumulated metabolites. The result of the metabolic pathway was analyzed by MetaboAnalyst 5.0 [82].

### 4.7. Combined Analysis of Metabolomics and Network Pharmacology

The common pathways from network pharmacology and metabolomics were selected by ggVenn. Joint pathway analysis with pathways, targets and metabolites was performed by Cytoscape3.9.0 software.

### 4.8. Molecular Docking

The operation procedure of molecular docking was performed as Kalirajan [83]. The RSCBPDB database [84], AlphaFold protein structure database [85], and PubChem database [86] were used to download protein ACSL4 (AF-Q9QUJ7-F1), PTGS2 (PDB ID: 1CVU), ALOX15 (AF-P39654-F1) and (PDB ID: 5L71GPX4) in the PDB format, and the 3D structures of puerarin, respectively. Maestro 11.9 of the Schrödinger suite 2019 was used for molecular docking. Firstly, the protein preparation wizard of the Schrödinger suite 2019 was used for preparing protein receptors, including refining bond orders, adding hydrogens, deleting water molecules beyond 5 Å, repairing missing chains and minimizing the energy of protein at the OPLS3e molecular force field. Secondly, the ligand preparation wizard was used to prepare the puerarin ligand. Subsequently, the grid boxes were generated by using the receptor grid generation module. The glide module of the Schrödinger suite 2019-4 in extra precision (XP) mode was used to dock the puerarin ligand with potential protein receptors and record the docking scores. According to the docking scores, we selected the best one (with the lowest docking scores) for the further prediction of binding energy. Finally, the free energy of binding for the puerarin ligand with potential protein receptors was predicted by the prime molecular mechanics-generalized Born surface area (MM-GB/SA) of Schrödinger 2019. All other docking parameters were set to the default values of the software.

### 4.9. Statistical Analysis

Data were analyzed with the GraphPad Prism 9.0 software (California, USA) and R, represented by mean ± standard deviation (SD) (n = 3). Data in all the bio-assays were evaluated by Student’s t-test or ANOVA followed by post-analysis.

## 5. Conclusions

In this study, we determined puerarin could play an anti-inflammatory role in RAW264.7. Then, the result of the network pharmacology, metabolomics, and molecular docking showed that puerarin counters inflammation mainly by affecting three ferroptosis-related pathways (arachidonic acid metabolism, tryptophan metabolism and glutathione metabolism), six metabolites (20-HETE, serotonin, kynurenine, oxidized glutathione, gamma-glutamylcysteine and cysteinylglycine) and four related targets (ACSL4, PTGS2, ALOX15 and GPX4). Hence, we suggested that one of the anti-inflammatory mechanisms of puerarin might be via regulating ferroptosis. Our present findings will enrich the anti-inflammatory molecular mechanism of puerarin, although further detailed research about the relationships between the anti-inflammatory effect and anti-ferroptosis effect of puerarin is still needed.

## Figures and Tables

**Figure 1 metabolites-12-00653-f001:**
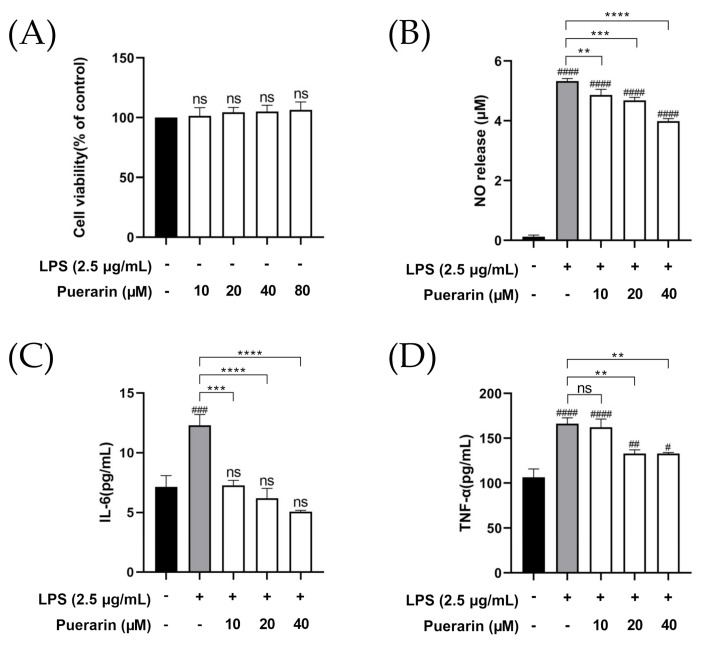
Puerarin-alleviated inflammation of LPS-induced RAW264.7 cells. (**A**) The cell viability; (**B**) the expression of NO, (**C**) IL-6 and (**D**) TNF-α in LPS-induced RAW264.7 cells when different concentrations of puerarin are incubated. (^#^
*p* < 0.05, ^##^
*p* < 0.01, ^###^
*p* < 0.001 and ^####^
*p* < 0.0001, compared with the control group; ** *p* < 0.01, *** *p* < 0.001 and **** *p* < 0.0001, compared with the LPS group; ns means not significant.).

**Figure 2 metabolites-12-00653-f002:**
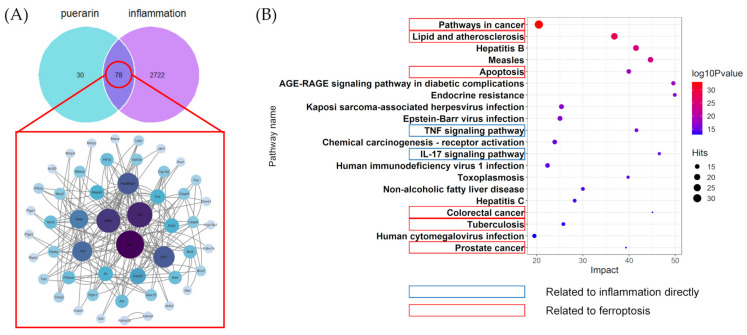
The network pharmacology analysis of puerarin anti-inflammatory. (**A**) The potential targets of puerarin countering inflammation and PPI network of 51 targets. (**B**) Bubble diagram of the top20 KEGG pathways.

**Figure 3 metabolites-12-00653-f003:**
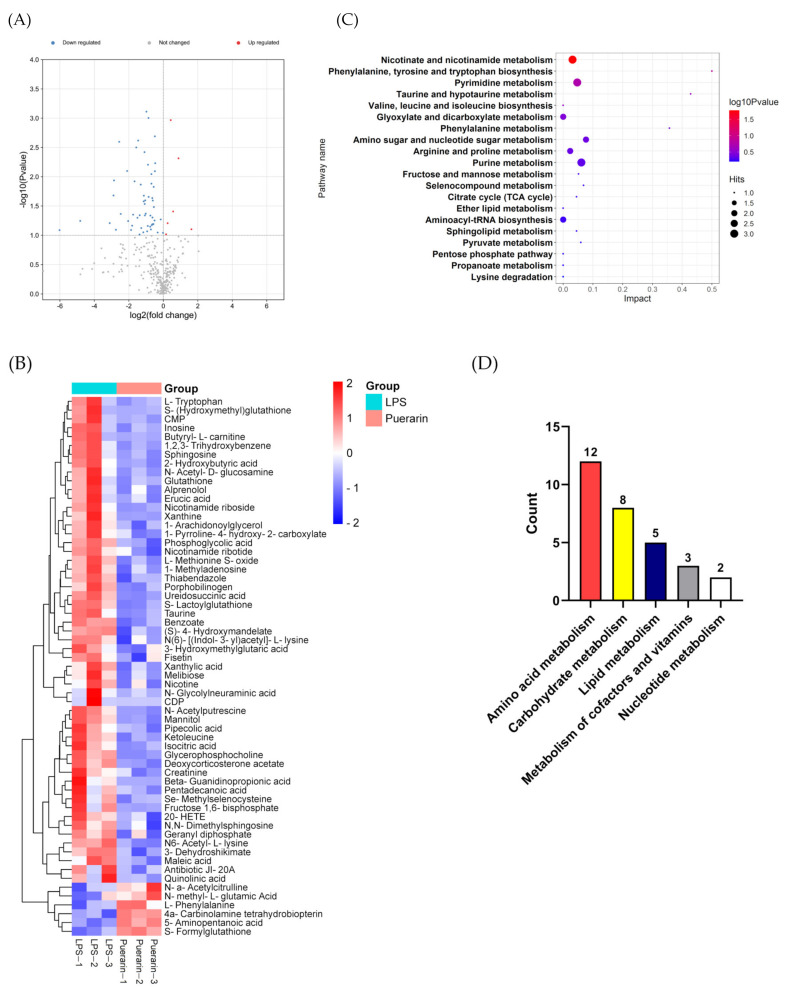
The Metabonomic analysis of puerarin anti-inflammatory. (**A**) Volcano figures and (**B**) heatmap of differential metabolites between the puerarin group and the LPS group. (**C**) Bubble diagram of the top20 KEGG pathways. (**D**) The number of KEGG pathways in each category.

**Figure 4 metabolites-12-00653-f004:**
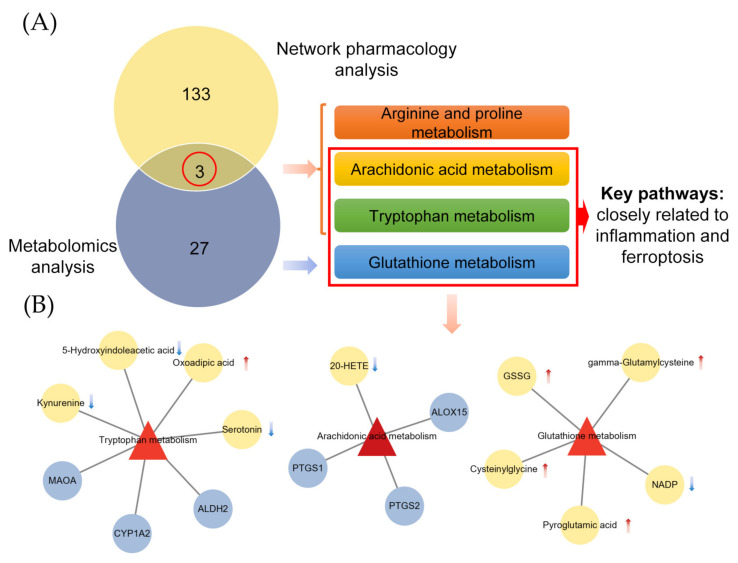
Combined analysis of metabolomics and network pharmacology. (**A**) The pathways co-enriched by metabolomics analysis and network pharmacology analysis. (**B**) Targets and metabolites enriched in key pathways (red triangles represent pathways, gray circles represent targets, yellow circles represent metabolites, red arrows represent up-regulated and blue arrows represent down-regulated).

**Figure 5 metabolites-12-00653-f005:**
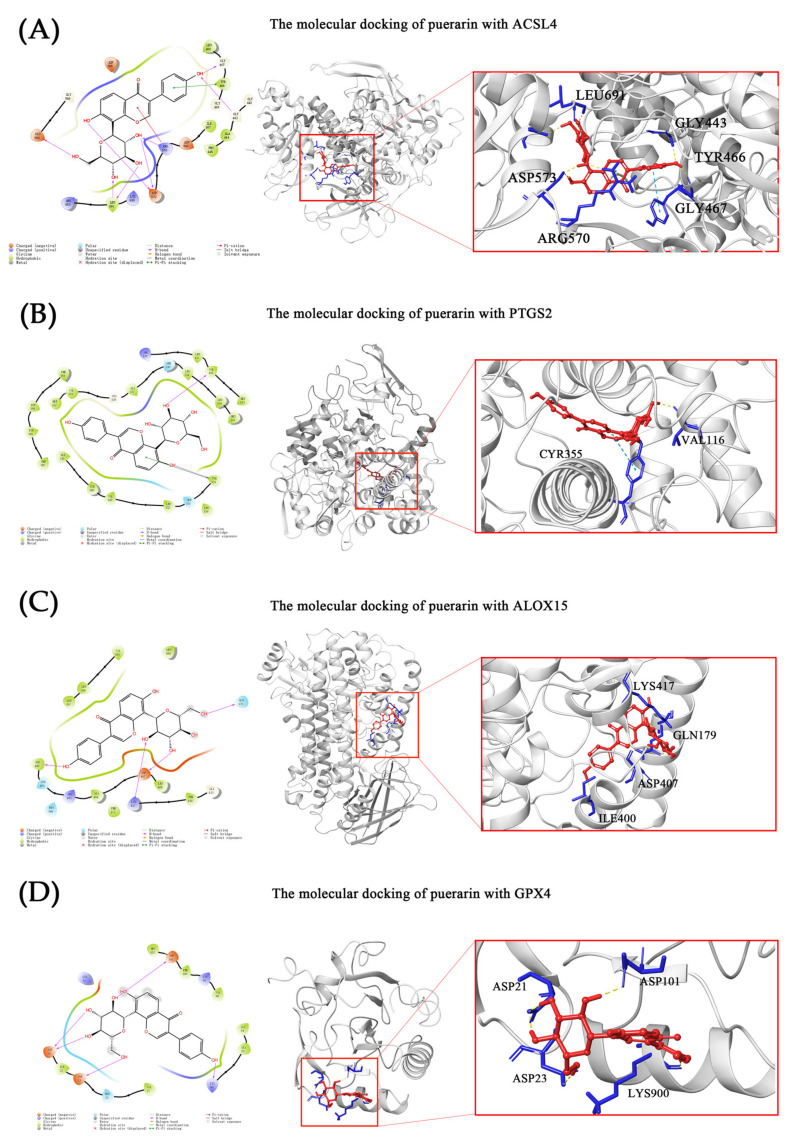
Predicted binding–interaction analysis and binding pose of (**A**) ACLS4, (**B**) PTGS2, (**C**) ALOX15 and (**D**) GPX4, with puerarin.

**Figure 6 metabolites-12-00653-f006:**
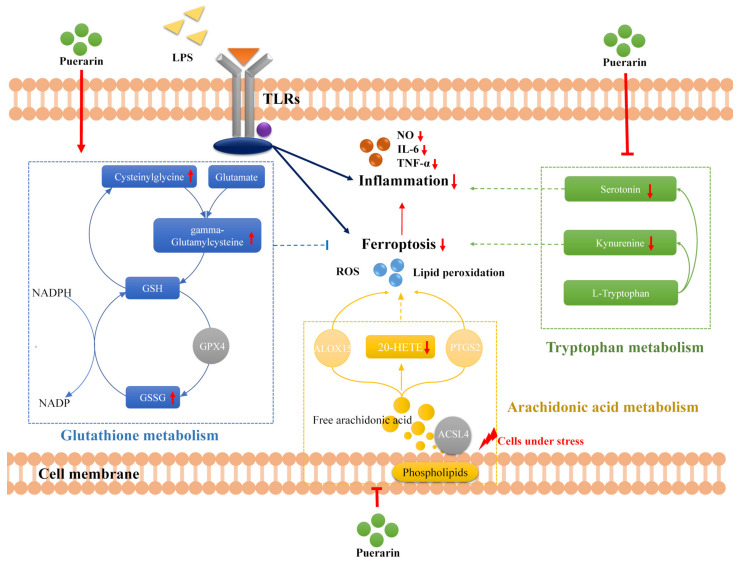
Hypothetical anti-inflammatory molecular mechanisms of puerarin by regulating ferroptosis. Rectangles represent metabolites and circles represent enzymes. Gray circles represent the biomarkers of ferroptosis. Lines with arrow-heads represent activation and lines with bars at the end denote inhibition.

**Table 1 metabolites-12-00653-t001:** The 18 common protein targets information of puerarin anti-inflammatory.

Gene Symbol	Uniprot ID	Protein Name	Degree
*Akt1*	P31750	RAC-alpha serine/threonine-protein kinase	30
*Jun*	P05627	Transcription factor Jun	26
*Stat3*	P42227	Signal transducer and activator of transcription 3	24
*Esr1*	P19785	Estrogen receptor	20
*Hsp90aa1*	P07901	Heat shock protein HSP 90-alpha	20
*Tnf*	P06804	Tumor necrosis factor	18
*Rela*	Q04207	Transcription factor p65	16
*Casp3*	P70677	Caspase-3	12
*Ar*	P19091	Androgen receptor	10
*Mapk9*	Q9WTU6	Mitogen-activated protein kinase 9	10
*Fos*	P01101	Protein c-Fos	10
*Esr2*	O08537	Estrogen receptor beta	10
*Ahr*	P30561	Aryl hydrocarbon receptor	8
*Bcl2*	P10417	Apoptosis regulator Bcl-2	8
*Hif1a*	Q61221	Hypoxia-inducible factor 1-alpha	8
*Bad*	Q61337	Bcl2-associated agonist of cell death	8
*Nos3*	P70313	Nitric oxide synthase, endothelial	8
*Prkaca*	P05132	cAMP-dependent protein kinase catalytic subunit alpha	8

**Table 2 metabolites-12-00653-t002:** The scores of Gilde docking and binding free energy.

Proteins	Interaction			Gilde Gscore (kcal/mol)	ΔG (kcal/mol)
	H-Bond	Pi–Pi Stacking	Pi–cation		
ACSL4	GLY443, GLY467, ARG570, ASP 573, LEU691 and GLU589	TYR466	ARG570	−7.808	−55.41
PTGS2	VAL116, TYR355	TYR355	-	−8.414	−24.62
ALOX15	GLN179, ILE400, ASP407 and LYS417	-	-	−6.858	−27.73
GPX4	ASP21, ASP23, LYS90 and ASP101	-	-	−4.872	−17.37

## Data Availability

All data are contained in the article and Appendix A.

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
