# Peer review of "Puerarin Induces Molecular Details of Ferroptosis-Associated Anti-Inflammatory on RAW264.7 Macrophages"

_metabolites, 2022, doi:10.3390/metabo12070653_

Round 1

Reviewer 1 Report

Abstract

The abstract is not well arranged. 

Puerarin is a natural flavonoid with significant anti-inflammatory effects, whose anti-inflammatory molecular mechanism has not been well clarified.

 However, Puerarin (P) has been studied extensively for its anti-inflammatory properties. Although, ferroptosis-mediated anti-inflammatory studies are not carried out yet. 

Introduction

Gene names are not written in the proper way. 

The former part seems reasonable- the authors introduced the functional component, the mechanism they desired to study- the latter part should be reconsidered. Especially the last paragraph.

Abbreviations should be checked for usage.

Results

I do not understand why the author chooses to use a conc. upon to 40, since 80 uM did not show reducibility in cell proliferation rate. 

Absolute NO release is missing.

Even though significance exists between 40uM and LPS only, 40uM P fails to reduce NO. Meanwhile, the author seems to omit the comparison between Control and other groups.

Table 1 check gene names

M&M

For Raw264.7 cells,  1×10/ well for 96 well plates seems sparse.

LPS/P supplementation procedure is missing for assays other then CCK8 assay.

The operation procedure of molecular docking was performed as Kalirajan [69]. 325 RSCBPDB (https://www.omicshare.com) database and PubChem database (https://pub- 326 chem.ncbi.nlm.nih.gov/) were used to download protein (ACSL4, PTGS2, ALOX15 and 327 GPX4) in PDB format, and the 3D structures of puerarin, respectively. Schrödinger suite 328 2019 was used for molecular docking.

The authors did not make citations in this part. PDB id should be added. The detail of docking parameters must be provided. The detailed interaction modes are not elucidated. 

For other assays, the replicates number is missing.

The discussion is not well written. Please discuss YOUR results with the potential mechanisms, while comparing yours with others. In the current version of ms, the authors separated their results and discussion in a confusing way. 

Figure 6. LPS could not make interaction without receptors. The illustration lacks scientific soundness.

Reviewer 2 Report

In this study, the authors intended to evaluate the effects of puerarin on ferroptosis associated anti-inflammatory on RAW264.7 Macrophages. Network pharmacology and metabolomics analysis were used to clarify the possible anti-inflammatory mechanism related to ferroptosis of puerarin. The key proteins associated with ferroptosis were bound to puerarin by molecular docking. The presented data are interesting and convincing. Some suggestion is listed below for the revision.

1.          In the study, the authors suggested that puerarin could inhibit ferroptosis as it might directly up-regulate anti-inflammation and anti-oxidative stress-related proteins. Do the authors have any idea on how puerarin crosses the cell membrane to react with these proteins?

2.          As shown in Figure1 A, the cell viability of puerarin, did the authors detect the cell viability while cells were exposed to puerarin and then with LPS? Furthermore, please label the dose of LPS in Figure 1.

3.          Please correct the words “whereupon” on lane 60, in which the letter “w” should be capitalized.

4.          Are there any other compounds (similar or different chemical structures) possessing comparable function like puerarin. Please provide some discussion.

Round 2

Reviewer 1 Report

Thank you for your great effort. 

The quality of the manuscript is improved.

Some minor opinions:

Check the figure caption and illustration.

e.g. Figure 4 (A) - network (of?) pharmacology  
